# Improving Uncertainty Quantification in Large Language Models via Semantic Embeddings

## Abstract

Hallucinations remain a major safety bottleneck for large language models (LLMs), necessitating effective detection methods such as quantifying uncertainty in the model's generations. While traditional uncertainty measures based on token likelihoods fail to capture semantic uncertainty, recent approaches like Semantic Entropy (SE) and Kernel Language Entropy (KLE) focus on isolating the underlying semantic uncertainty of the LLM. However, these methods impose significant computational overhead beyond generating samples: they require numerous natural language inference (NLI) calls to compare outputs, limiting their use in latency-sensitive applications. We introduce **Semantic Embedding Uncertainty (SEU)**, a lightweight metric that directly measures semantic disagreement in embedding space. Like SE and KLE, SEU requires multiple model outputs, but crucially simplifies the subsequent analysis. SEU computes uncertainty as the average pairwise cosine distance between sentence embeddings—requiring only $M$ embedding model forward passes followed by $O(M^2)$ dot products, instead of $O(M^2)$ NLI forward passes. SEU thus facilitates real-time semantic uncertainty quantification in applications where latency is paramount. Experiments on question answering and reasoning tasks demonstrate that SEU achieves comparable or superior accuracy to SE and KLE while reducing inference latency by up to 100x, enabling its deployment in resource-constrained settings.

## 1 Introduction

Large language models have transformed the landscape of artificial intelligence, demonstrating remarkable capabilities in code generation, medical record analysis, and natural conversation (The Gemini Team,, 2023; Touvron et al., 2023a; OpenAI, 2023; Brown et al., 2020). Yet beneath their fluent outputs lies a critical vulnerability: these models routinely generate plausible-sounding information that is entirely fabricated, a phenomenon known as hallucination (Ji et al., 2023; Filippova, 2020; Maynez et al., 2020; Tian et al., 2024). When a medical AI assistant confidently states incorrect drug interactions or a legal chatbot cites non-existent case law, the consequences extend far beyond mere inconvenience. As LLMs increasingly support high-stakes decisions in healthcare (Singhal et al., 2023), finance, and legal practice (Weiser, 2023), our ability to detect when these systems venture beyond their knowledge becomes paramount for safe deployment.

Traditional uncertainty quantification fails for natural language. Classical approaches like Bayesian methods (Wilson & Izmailov, 2020), ensembles (Lakshminarayanan et al., 2017), and Monte Carlo dropout (Gal & Ghahramani, 2016) measure confidence through token probabilities. But these probabilities conflate two fundamentally different uncertainties: what to say (semantic) versus how to say it (lexical). Consider this problem: A model answering "What is the capital of France?" assigns equal probability to "Paris is the capital" and "The capital city is Paris". Traditional metrics interpret this probability mass spread as uncertainty, yet the model knows the answer perfectly and such uncertainty is simply due to the phrasing. Meanwhile, a model confidently generating "Lyon is the capital of France" appears certain. This conflation becomes particularly problematic for hallucination detection as a model generating multiple paraphrases of a correct fact can appear uncertain, while one confidently producing a single fabrication appears certain. For a vocabulary size $|\mathcal{T}|$ and sentences of length $N$, computing exact semantic uncertainty would require

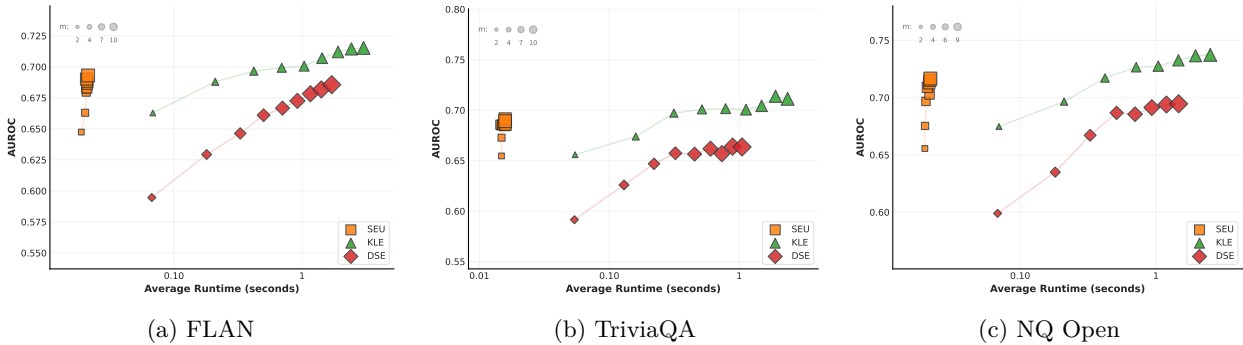

(a) FLAN

(b) TriviaQA

(c) NQ Open

Figure 1: Hallucination detection performance (AUROC) versus computational runtime trade-off for semantic uncertainty methods across three question-answering datasets. Each subplot shows AUROC scores plotted against average runtime for post-processing the generated responses, with varying numbers of sampled responses (m = 2, 4, 6,.. shown as different points). As the number of samples (M) increases, semantic methods generally show improved AUROC performance at the cost of increased post processing runtime, with runtime scaling particularly steeply for NLI-based methods. SEU (Semantic Embedding Uncertainty) achieves comparable AUROC to state-of-the-art KLE (Kernel Language Entropy) while operating approximately 100× faster. DSE (Discrete Semantic Entropy) shows intermediate performance and runtime. Results are averaged across six language models (Gemma 2-9B/27B, Llama 3-8B/70B, Mistral 7B, Mixtral 8×7B).

integrating over $|\mathcal{T}|^N$ possible sequences, which is an intractable problem that forces us to use approximations (Kuhn et al., 2023).

Semantic Entropy (SE) (Kuhn et al., 2023) introduced a simple idea to estimate semantic uncertainty: sample multiple complete responses, then cluster them by meaning. Specifically, SE groups responses into semantic clusters by checking bidirectional entailment, ensuring that paraphrases are merged into a single cluster while distinct meanings form separate ones. By computing entropy over these semantic clusters rather than tokens, SE measures semantic uncertainty.

While SE requires access to token probabilities to weight clusters, Discrete Semantic Entropy (DSE) (Farquhar et al., 2024a) adapts this approach for black-box settings by treating each sampled generation as equally likely. DSE uses the same bidirectional entailment clustering as SE but approximates cluster probabilities using empirical frequencies. This modification enables uncertainty quantification for closed-source APIs where internal probabilities are inaccessible, though DSE still inherits SE's computational bottleneck of requiring $O(M^2)$ NLI comparisons for clustering.

However, SE's binary clustering based on a strict entailment criterion can miss nuances. For example, it treats "discovered by Kolmogorov" and "discovered by Laplace" as equally different from "discovered by Shakespeare" when clearly the first two are more related. Kernel Language Entropy (KLE) fixed this by replacing binary clustering with continuous similarity scores (Nikitin et al., 2024). Using natural language inference (NLI) model probabilities to build a semantic similarity matrix, KLE computes von Neumann entropy over continuous representations. This change provides more nuanced comparisions between responses and achieves state-of-the-art performance across benchmarks.

Despite their empirical success, semantic uncertainty methods face a significant computational barrier. The computational cost stems from their reliance on pairwise semantic comparisons: for $M$ sampled responses, both SE and KLE require $O(M^2)$ forward passes through an NLI model to assess whether each pair exhibits bi-directional entailment. In deployments with $M = 10$ samples, this translates to up to nearly a hundred NLI model forward passes. Our experiments show these operations push total runtime into multi-second territory which can be prohibitive for applications expecting sub-second responses.

The fundamental question thus becomes: can we design a semantic uncertainty method that satisfies three critical requirements: First, it must maintain **black-box compatibility**, working with API-only models where internal probabilities are inaccessible, which is a key strength inherited from DSE and KLE. Second, it

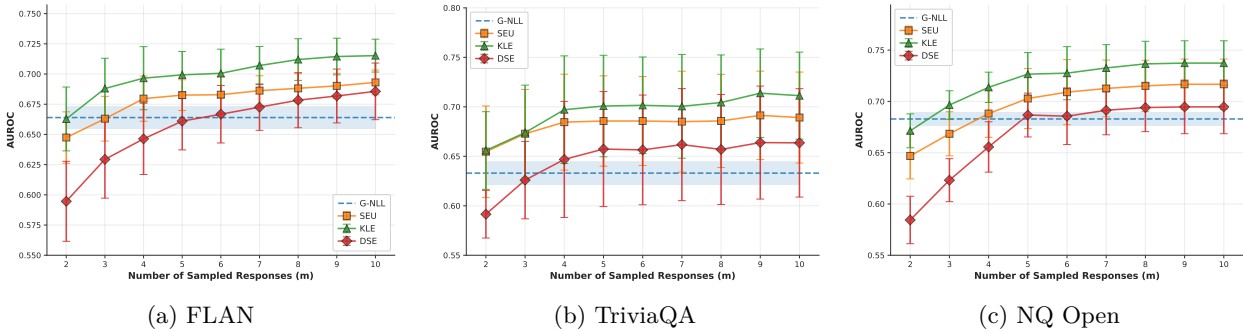

Figure 2: Hallucination detection performance (AUROC) as a function of the number of sampled responses ($M$) for different uncertainty quantification methods, averaged across six language models. SEU maintains competitive performance with KLE across varying sample sizes while requiring significantly less computation. The performance gap between semantic methods (SEU, KLE, DSE) and token-level methods (G-NLL) widens as more samples are used, highlighting the importance of semantic-level uncertainty quantification.

should preserve KLE's continuous semantic representation, which demonstrably improves uncertainty quantification by capturing fine-grained semantic relationships rather than forcing binary equivalence decisions. Third, and crucially, it must eliminate the computational bottleneck of $O(M^2)$ NLI forward passes that both DSE and KLE require for computing bidirectional entailment between every pair of responses.

The key insight lies in recognising that the geometric structure of embedding space already encodes the semantic relationships that KLE use a NLI model to compute. Modern sentence embeddings place semantically similar text nearby in vector space. When KLE uses NLI comparisons to determine that "discovered by Kolmogorov" relates more closely to "discovered by Laplace" than to "discovered by Shakespeare", it is estimating the structure that embeddings can provide instantly.

To exploit this observation, we propose Semantic Embedding Uncertainty (SEU), a lightweight metric that directly leverages embedding space for semantic uncertainty quantification. Rather than computing bidirectional entailment between all response pairs, which requires $O(M^2)$ NLI forward passes, SEU embeds each response once and computes pairwise cosine similarities. This replaces expensive neural network forward passes with simple linear algebra, requiring $M$ embedding computations plus $O(M^2)$ **dot products** that modern hardware accelerates through parallelisation. SEU maintains all three critical requirements: black-box compatibility (only needs text outputs), continuous semantic representation (distances capture degrees of similarity), and practical runtime (milliseconds instead of seconds). By recognising that embeddings already solve the semantic similarity problem, SEU delivers the benefits of semantic uncertainty at significantly faster speeds.

## 2 Background

### 2.1 Semantic Entropy

Semantic Entropy (SE) (Kuhn et al., 2023) aims to distinguish semantic uncertainty from lexical variation. Given input $x$, SE samples $M$ responses $s^{(1)}, s^{(2)}, \ldots, s^{(M)}$ using temperature sampling Holtzman et al. (2020) and records their probabilities $P(s^{(m)}|x)$ when available. The key idea lies in clustering responses by meaning rather than analysing raw token probabilities. SE groups responses through bidirectional entailment: responses $s_i$ and $s_j$ share a cluster if and only if $s_i$ entails $s_j$ AND $s_j$ entails $s_i$, as determined by an NLI model (e.g., DeBERTa-Large-MNLI(He et al., 2021)). The clustering proceeds greedily, each response either joins an existing cluster if it bidirectionally entails any member, or forms a new cluster. This creates the computational bottleneck: determining cluster membership requires $O(M^2)$ NLI forward passes. SE computes entropy over the resulting semantic clusters. Each cluster's probability equals the sum of its members' probabilities: $P(C_i|x) = \sum_{s_j \in C_i} P(s_j|x)$. The final uncertainty score is the Shannon entropy over

these clusters: $SE(x) = -\sum_i P(C_i|x) \log P(C_i|x)$. High entropy signals responses spread across multiple semantic meanings, indicating potential hallucination.

For API-only models without access to $P(s|x)$, Discrete Semantic Entropy (DSE) (Farquhar et al., 2024b) approximates cluster probabilities using empirical frequencies: $P(C_i|x) \approx |C_i|/M$. While this enables semantic uncertainty for closed-source models, DSE still requires the same $O(M^2)$ NLI comparisons which is the primary computational barrier for real-time deployment.

## 2.2 Kernel Language Entropy

Kernel Language Entropy (KLE) (Nikitin et al., 2024) extended SE by replacing binary clustering with continuous semantic similarity. Given input $x$, KLE samples $M$ responses $s^{(1)}, s^{(2)}, ..., s^{(M)}$ and constructs a weighted semantic graph $G_{\text{sem}}$ where nodes represent responses and edge weights encode semantic similarity. To compute edge weights, KLE uses an NLI model (e.g., DeBERTa) that outputs probabilities for three classes: entailment, neutral, and contradiction. The weight $W_{ij}$ between responses $s^{(i)}$ and $s^{(j)}$ is a weighted combination of both entailment and neutrality probabilities in both directions: $W_{ij} = f(\text{NLI}(s^{(i)}, s^{(j)}), \text{NLI}(s^{(j)}, s^{(i)}))$. This creates a soft similarity score rather than SE's binary equivalence. Critically, KLE must compute all $M^2$ pairwise weights to construct the kernel matrix. While DSE can terminate comparisons once cluster membership is determined, KLE requires every pairwise NLI evaluation upfront.

From the weight matrix $W$, KLE derives a kernel matrix $K_{sem}$ using graph kernels like Heat Kernel (Kondor & Lafferty, 2002) that capture semantic structure. The final uncertainty score is the von Neumann entropy of the kernel matrix. This continuous approach uses nuanced semantic relationships, recognising that "discovered by Kolmogorov" relates more closely to "discovered by Laplace" than to "discovered by Shakespeare." However, the mandatory $O(M^2)$ NLI forward passes make KLE even more computationally expensive than DSE, which can exploit early cluster assignment to reduce comparisons in practice.

# 3 Methods

## 3.1 Semantic Embedding Uncertainty

Both SE and KLE require $O(M^2)$ NLI forward passes—a computational barrier that prevents real-time deployment. Yet KLE demonstrates that continuous semantic similarity significantly improves uncertainty quantification over binary clustering. The question becomes: can we preserve KLE's continuous semantics while eliminating its computational bottleneck? Modern sentence embeddings already encode the semantic similarity that KLE laboriously computes. When embedding models like Sentence-BERT (Reimers & Gurevych, 2019) map text to vectors, they place semantically similar sentences nearby in embedding space. Thus, two paraphrases land close together; contradictory statements sit far apart. This spatial structure captures precisely the continuous semantic relationships that KLE extracts through expensive NLI comparisons.

We propose Semantic Embedding Uncertainty (SEU), which directly exploits this geometric structure. Rather than computing $M^2$ NLI forward passes to build a similarity matrix, SEU simply embeds each response once (M forward passes through an embedding model) and measures distances in embedding space. The computational gains are significant because we reduce neural network forward passes from $O(M^2)$ to $O(M)$, and replace the remaining comparisons with fast vector arithmetic that modern hardware accelerates efficiently.

The SEU algorithm works as follows. Given an input query $x$, we sample $M$ output sequences $s^{(1)}, s^{(2)}, \ldots, s^{(M)}$ from the language model's predictive distribution $p(s|x)$. We then encode each sequence using a pretrained sentence embedding model $\phi$, yielding embeddings $e^{(1)}, e^{(2)}, \ldots, e^{(M)}$ where $e^{(i)} = \phi(s^{(i)})$. Finally, we quantify semantic uncertainty as one minus the average pairwise cosine similarity:

$$\text{SEU}(x) = 1 - \frac{2}{M(M-1)} \sum_{i=1}^{M-1} \sum_{j=i+1}^{M} \cos\left(\mathbf{e}_i, \mathbf{e}_j\right), \tag{1}$$

where $\cos\left(\mathbf{e}_i, \mathbf{e}_j\right) = \frac{\mathbf{e}_i \cdot \mathbf{e}_j}{\|\mathbf{e}_i\| \|\mathbf{e}_j\|}$ denotes the cosine similarity between embedding vectors $\mathbf{e}_i$ and $\mathbf{e}_j$. This formulation ensures SEU ranges from 0 (perfect agreement) to approximately 1 (maximal disagreement), providing an interpretable uncertainty score.

### 3.2 Design Rationale

#### 3.2.1 Why Cosine Similarity

Modern sentence embedders like SimCSE (Gao et al., 2021) and NV-EMBED (Lee et al., 2025) are trained specifically to optimise cosine similarity often through contrastive learning. When these models see "Paris is the capital of France" and "France's capital city is Paris", they map them to nearly parallel vectors. This training objective directly aligns with our goal of measuring semantic agreement. Furthermore, cosine similarity is a dominant metric for semantic comparison benchmarks. The MTEB benchmark (Muennighoff et al., 2022) which forms the standard for evaluating embedding models uses cosine similarity as one of its primary metric across datasets.

The embedding-based approach offers a significant computational advantage over NLI-based methods. While SE and KLE require $O(M^2)$ forward passes through an NLI model, SEU needs only $M$ forward passes through an embedding model followed by $\frac{M(M-1)}{2}$ dot product operations. These dot products execute efficiently on modern hardware through vectorised operations, reducing the computational bottleneck from NLI model inference to simple dot products.

SEU assumes that the embedding model provides a meaningful semantic neighbourhood structure for the target domain, i.e., that embedding cosine similarity correlates with semantic equivalence among LLM responses. When this assumption holds, SEU approximates the pairwise semantic similarity that KLE computes via NLI inference. We validate this assumption empirically in Appendix A.4, showing strong agreement between embedding similarity and NLI-based similarity (Spearman $\rho = 0.86$, Pearson $r = 0.75$). Practitioners can verify applicability to new domains by computing this correlation on a small held-out sample; the MTEB leaderboard (Muennighoff et al., 2022) also provides domain-level quality scores to guide embedding model selection.

SEU's modular architecture offers a potential advantage for domain adaptation: the embedding model can be swapped without retraining the UQ pipeline. For instance, domain-specific embeddings such as BioBERT (Lee et al., 2019) or ClinicalBERT (Alsentzer et al., 2019) for medical text, FinBERT (Araci, 2019) for finance, and LEGAL-BERT for legal settings could serve as drop-in replacements. We note that this remains an architectural affordance rather than a demonstrated capability; empirical validation in specialised domains is left for future work.

## 4 Experiments

We evaluate SEU against strong established uncertainty quantification baselines. Our comparison includes greedy negative log-likelihood (G-NLL), which has been shown to outperform NLL and SE in some cases, as a single-sample baseline (Aichberger et al., 2024), Discrete Semantic Entropy (DSE) for black-box evaluation (Farquhar et al., 2024b), Kernel Language Entropy (KLE) as the current state-of-the-art (Nikitin et al., 2024), and our proposed SEU. We follow these prior works and measure each method's ability to detect hallucinations using area under the receiver operating characteristic curve (AUROC).

Our experiments span six language models: Gemma 2 (9B and 27B) (Gemma Team, 2024), Llama 3 (8B and 70B) (Touvron et al., 2023b), Mistral 7B (Jiang et al., 2023), and Mixtral 8×7B (Jiang et al., 2024). We use instruction tuned versions of these models. This selection covers both dense and mixture-of-experts

Table 1: Hallucination detection performance (AUROC) across four question-answering datasets and six language models. Values show mean and standard error over multiple random seeds. SEU achieves performance within 2-3 points of KLE while running 100× faster.

| Dataset | Model | G-NLL | SEU | KLE | DSE |
|---|---|---|---|---|---|
| FLAN | Gemma 9B | $0.682 \pm 0.017$ | $0.710 \pm 0.014$ | $0.715 \pm 0.028$ | $0.689 \pm 0.026$ |
| | Gemma 27B | $0.648 \pm 0.019$ | $0.698 \pm 0.001$ | $0.718 \pm 0.009$ | $0.689 \pm 0.011$ |
| | Llama 8B | $0.730 \pm 0.010$ | $0.678 \pm 0.005$ | $0.701 \pm 0.011$ | $0.656 \pm 0.013$ |
| | Llama 70B | $0.620 \pm 0.025$ | $0.671 \pm 0.025$ | $0.713 \pm 0.015$ | $0.652 \pm 0.014$ |
| | Mistral 7B | $0.674 \pm 0.022$ | $0.690 \pm 0.011$ | $0.737 \pm 0.016$ | $0.697 \pm 0.017$ |
| | Mixtral 24B | $0.602 \pm 0.019$ | $0.651 \pm 0.005$ | $0.674 \pm 0.015$ | $0.622 \pm 0.016$ |
| MATH | Gemma 9B | $0.732 \pm 0.014$ | $0.694 \pm 0.004$ | $0.690 \pm 0.023$ | $0.509 \pm 0.018$ |
| | Gemma 27B | $0.681 \pm 0.005$ | $0.712 \pm 0.017$ | $0.668 \pm 0.005$ | $0.469 \pm 0.004$ |
| | Llama 8B | $0.595 \pm 0.025$ | $0.629 \pm 0.048$ | $0.592 \pm 0.008$ | $0.491 \pm 0.007$ |
| | Llama 70B | $0.517 \pm 0.018$ | $0.641 \pm 0.007$ | $0.651 \pm 0.002$ | $0.559 \pm 0.004$ |
| | Mistral 7B | $0.602 \pm 0.003$ | $0.543 \pm 0.019$ | $0.572 \pm 0.019$ | $0.514 \pm 0.010$ |
| TriviaQA | Gemma 9B | $0.640 \pm 0.010$ | $0.702 \pm 0.025$ | $0.712 \pm 0.029$ | $0.697 \pm 0.026$ |
| | Gemma 27B | $0.583 \pm 0.017$ | $0.620 \pm 0.008$ | $0.624 \pm 0.012$ | $0.604 \pm 0.005$ |
| | Llama 8B | $0.679 \pm 0.018$ | $0.711 \pm 0.038$ | $0.747 \pm 0.040$ | $0.715 \pm 0.031$ |
| | Llama 70B | $0.610 \pm 0.040$ | $0.611 \pm 0.011$ | $0.650 \pm 0.018$ | $0.593 \pm 0.017$ |
| | Mistral 7B | $0.647 \pm 0.022$ | $0.730 \pm 0.008$ | $0.751 \pm 0.014$ | $0.702 \pm 0.008$ |
| | Mixtral 24B | $0.567 \pm 0.001$ | $0.632 \pm 0.032$ | $0.657 \pm 0.021$ | $0.566 \pm 0.032$ |
| NQ Open | Gemma 9B | $0.692 \pm 0.016$ | $0.700 \pm 0.019$ | $0.714 \pm 0.017$ | $0.681 \pm 0.018$ |
| | Gemma 27B | $0.678 \pm 0.002$ | $0.722 \pm 0.008$ | $0.722 \pm 0.011$ | $0.700 \pm 0.009$ |
| | Llama 8B | $0.732 \pm 0.013$ | $0.724 \pm 0.011$ | $0.730 \pm 0.029$ | $0.680 \pm 0.017$ |
| | Llama 70B | $0.691 \pm 0.014$ | $0.682 \pm 0.018$ | $0.710 \pm 0.022$ | $0.663 \pm 0.016$ |
| | Mistral 7B | $0.668 \pm 0.025$ | $0.705 \pm 0.027$ | $0.722 \pm 0.023$ | $0.692 \pm 0.024$ |
| | Mixtral 24B | $0.636 \pm 0.007$ | $0.675 \pm 0.021$ | $0.727 \pm 0.024$ | $0.680 \pm 0.015$ |

architectures, from efficient 7B models to powerful 70B models. This range ensures our findings generalise beyond specific model families or sizes.

For each query, we vary the number of responses $M \in \{2, ..., 10\}$ using temperature sampling (T=1) and encode them with bge-large-en-v1.5 (Xiao et al., 2023), a sentence transformer that balances efficiency with representation quality. We also compare it against Qwen3-Embedding-0.6B (Zhang et al., 2025) in figure 4. For SE and KLE we use DeBERTa-Large-MNLI (He et al., 2021), following their original implementations. We conduct timing experiments on a single RTX-A6000 GPU using mixed-precision inference, with full implementation details in Appendix A. This choice ensures that the NLI model and the embedding model are of comparable size (approximately 406 million and 335 million parameters, respectively).

Following the "LLM-as-a-judge" protocol, we evaluate answer correctness with the Llama 3.3-70B-Instruct model (Grattafiori et al., 2024). We prompt this judge using the exact rubric and templates proposed by Santilli et al. (2024), ensuring our evaluation is immune to the length–metric interaction they uncovered.

## 4.1 Datasets

We evaluate on four standard benchmarks that stress different aspects of model knowledge. Natural questions subset of FLAN (Wei et al., 2022) tests general instruction-following across diverse tasks. MATH-500 (Hendrycks et al., 2021) challenges mathematical reasoning where hallucinations often manifest as incorrect computational steps. TriviaQA (Joshi et al., 2017) probes factual knowledge across broad domains. Natural Questions Open (NQ Open) (Kwiatkowski et al., 2019) also requires retrieving specific facts from the model's parametric memory. Together, these datasets provide comprehensive coverage of scenarios where hallucination detection matters most.

## 4.2 Main Results

To evaluate SEU comprehensively, we address three key questions: (1) Does SEU achieve comparable hallucination detection performance to state-of-the-art methods like KLE while offering superior computational efficiency? (2) How does SEU's performance vary across different datasets and language models? (3) How sensitive are semantic uncertainty methods, including SEU, to experimental parameters such as the number of sampled responses ($M$) or the choice of embedding model?

**Efficiency Analysis.** Figure 1 presents our core findings on the trade-off between detection accuracy (AUROC) and computational runtime. While KLE achieves high AUROC scores, it incurs substantial runtime costs, often taking seconds per evaluation due to its reliance on numerous NLI forward passes. In contrast, SEU matches or closely approaches KLEs AUROC values while executing in milliseconds, delivering up to a 100x speed-up over SE and KLE. This positions SEU uniquely for latency-sensitive applications, enabling real-time semantic uncertainty quantification without compromising on effectiveness. Throughout our experiments, runtime refers to the *post-processing overhead* of each UQ method, i.e., the time to compute an uncertainty score from already-sampled responses. Sampling cost is shared across all methods and excluded from comparisons.

**Performance Variability Across Datasets and Models.** Table 1 details AUROC performance across six language models (Gemma 2, Llama 3, Mistral, and Mixtral) spanning 7B to 70B parameters, representing diverse training methodologies, tokenizers, and architectural choices. SEU demonstrates consistent performance across this model diversity, maintaining a small gap with KLE overall.

On the FLAN dataset, which encompasses a wide variety of task types, KLE holds a slight edge, outperforming SEU by 2-4 AUROC points across most models. Results on TriviaQA and NQ Open, both factual question-answering benchmarks, show KLE maintaining moderate advantages of 2-3 AUROC points, benefiting from its ability to detect fine-grained semantic distinctions.

However, on the MATH dataset, we observe an interesting reversal, SEU outperforms KLE for three of the five models tested. models tested. Mathematical responses, often containing symbolic expressions and structured reasoning, may not align well with NLI models trained primarily on natural language entailment. SEU's embedding-based approach proves more robust to these domain shifts, highlighting its strengths in specialised domains.

**Sensitivity to Experimental Setup.** Figure 2 examines the impact of varying the number of sampled responses $M$ from 2 to 10. Semantic methods (SEU, KLE, DSE) exhibit steady AUROC improvements with increasing $M$, as additional samples better reveal the model's semantic distribution. In contrast, G-NLL, relying on single-sample token probabilities, plateaus immediately. This underscores the need for larger $M$ in semantic methods to achieve peak performance, but it also amplifies computational costs—for KLE, scaling to $M = 10$ requires up to 100 NLI calls, forcing a trade-off between accuracy and latency. SEU mitigates this by making large $M$ computationally feasible, allowing robust uncertainty estimation without latency penalties.

Additionally, Figure 4 assesses sensitivity to embedding model choice, comparing bge-large-en-v1.5 ( 335M parameters) against Qwen3-Embedding-0.6B ( 600M parameters). SEU's performance remains stable across these encoders, indicating low sensitivity to the specific embedding model as long as it provides high-quality semantic representations.

## 5 Related Works

**Hallucinations in LLMs:** The challenge of hallucination detection in LLMs has become increasingly important as these models are deployed in real-world applications. Various benchmarks have been developed to evaluate this phenomenon, including TruthfulQA (Lin et al., 2021), FactualityPrompt (Lee et al., 2022), FActScore (Min et al., 2023), HaluEval (Li et al., 2023a), and FACTOR (Muhlgay et al., 2023). Early research on hallucinations primarily focused on issues in summarization tasks, where models would generate content unfaithful to the source text (Maynez et al., 2020; Durmus et al., 2020; Wang et al., 2020). This work laid the foundation for understanding the broader challenge of hallucinations in LLMs.

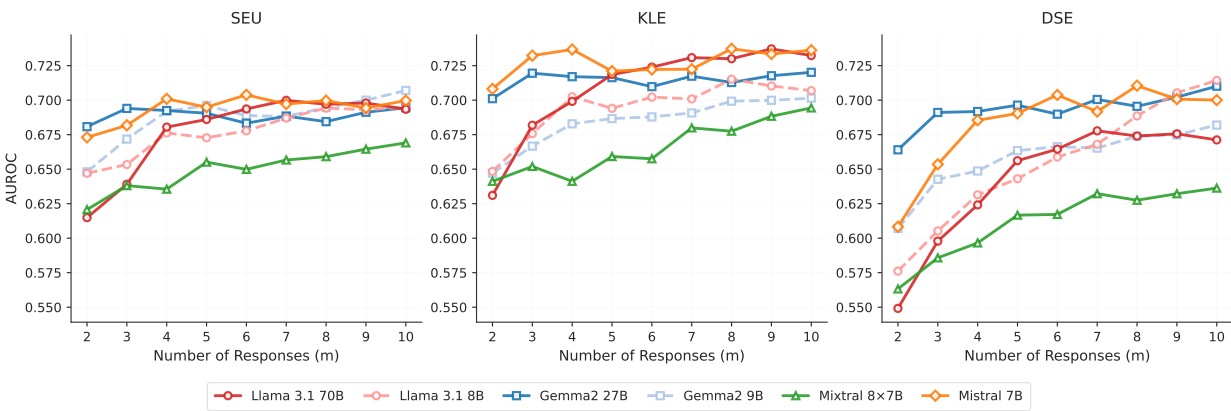

Figure 3: Hallucination detection performance (AUROC) across different language models on the FLAN dataset, comparing three semantic uncertainty methods. The figure shows: (a) SEU maintains competitive performance with KLE across all model scales while requiring much less computation, (b) performance generally improves with model scale for all methods, and (c) the gap between SEU and KLE remains relatively consistent ( 2-4 points) regardless of model size.

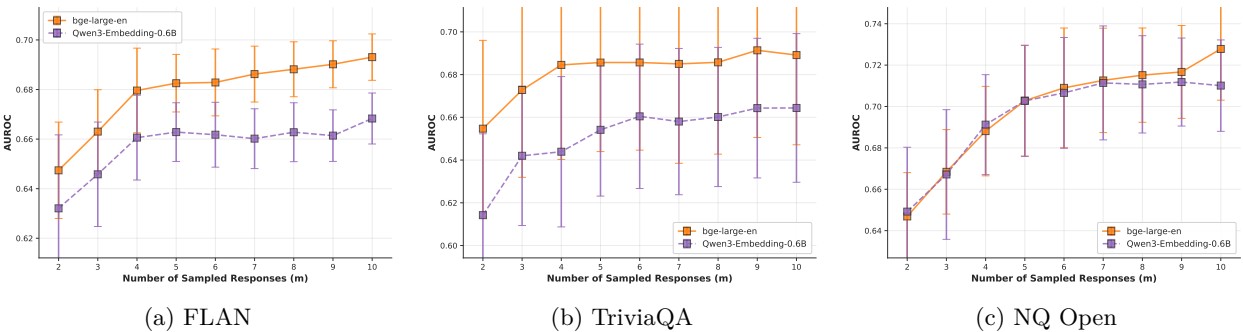

|  (a) FLAN | (b) TriviaQA | (c) NQ Open |

Figure 4: Impact of embedding model choice on SEU performance across three question-answering datasets. BGE-large-en-v1.5 achieves higher AUROC than Qwen3-Embedding-0.6B on TriviaQA and FLAN, while both models perform comparably on NQ Open. Results averaged across six language models.

**Uncertainty Estimation Approaches:** A significant body of work has explored methods to estimate uncertainty in LLM outputs. Many of these approaches rely on comparing multiple model generations or outputs by leveraging additional LLMs or by using the same LLM (Duan et al., 2023; Chen & Mueller, 2023; Manakul et al., 2023; Mündler et al., 2023). The field has seen a variety of innovative techniques, including those proposed by Kadavath et al. (2022), Mitchell et al. (2022), and Xu et al. (2022), which leverage different aspects of model behaviour to gauge uncertainty.

**Knowledge Integration Methods:** Another line of research focuses on integrating external knowledge to verify and improve the factual accuracy of LLM outputs. The RARR framework (Gao et al., 2023) uses search engines for knowledge retrieval and correction. Similarly, the Verify-and-Edit approach (Zhao et al., 2023) leverages external information sources. However, these methods face challenges in resolving conflicts between model knowledge and retrieved information, as highlighted by Shi et al. (2023). Additional work in this area includes efforts by Dziri et al. (2021), Peng et al. (2023), and Li et al. (2023c), who explore various techniques for grounding LLM outputs in external knowledge sources.

**Generation and Fine-tuning Strategies:** Researchers have also developed strategies to reduce hallucinations during the generation process or through model fine-tuning. Lee et al. (2022) introduced factual-nucleus sampling to balance output diversity and factual accuracy. Reinforcement learning from human feedback (RLHF) has been employed by Ouyang et al. (2022) and Touvron et al. (2023a) to align LLMs with desired

criteria, including truthfulness. Other approaches include careful curation of instruction-tuning data (Zhou et al., 2023) and linguistic calibration techniques (Mielke et al., 2022). Recent work by Tian et al. (2024) has further explored fine-tuning strategies specifically targeting factuality improvement.

**Leveraging the Latent Space:** An emerging area of research investigates the internal representations of LLMs to understand and manipulate their behaviour. Studies have suggested the existence of a "truthfulness" direction in the latent space of these models. For example, Li et al. (2023b) proposed Inference-Time Intervention to identify and modify factuality-related directions in model activations. Azaria & Mitchell (2023) introduced SAPLMA, suggesting that LLMs may have an internal awareness of their own inaccuracies. This line of inquiry has been further developed by Burns et al. (2023), who explored methods for discovering latent knowledge, and Marks & Tegmark (2023), who examined the geometry of truth representations in LLMs. Additional insights have been provided by Subramani et al. (2022) and Zou et al. (2023), who have explored techniques for understanding and manipulating the internal representations of these models. Kossen et al. (2024) further propose to directly predict the semantic entropy using probes acting on different hidden layers of the LLM.

While the above approaches tackle hallucination detection from various angles like external knowledge integration, model fine-tuning, or internal representations—they either require access to model internals, external resources, or expensive retraining. Our work specifically addresses the computational bottleneck in semantic uncertainty methods (SE, DSE, KLE), which have proven effective but require expensive NLI computations. Unlike methods that require model modifications, external knowledge bases, or access to internal states, SEU works immediately with any LLM API. By recognising that modern embeddings already encode the semantic similarities that existing methods laboriously compute through NLI, we achieve comparable uncertainty quantification at $100\times$ faster post-processing speed.

## 6 Conclusion

In this work, we introduced Semantic Embedding Uncertainty (SEU), a method that makes estimating semantic uncertainty quantification significantly faster than current baselines. By recognising that modern sentence embeddings already encode the semantic similarity that existing methods estimate through expensive NLI comparisons, SEU achieves comparable hallucination detection performance to state-of-the-art methods while operating $100\times$ faster. Our experiments across six language models and multiple datasets demonstrate that SEU maintains within a few AUROC points of KLE while reducing computational requirements from $O(M^2)$ NLI forward passes to $M$ embedding computations plus simple dot products. Because KLE and DSE operate at a multi-second latency, SEU by working on a milli-second latency can enable real-time hallucination detection in latency sensitive applications. Moreover, SEU's modular design means practitioners can swap in domain-specific embedding models without creating new NLI datasets, providing a potential pathway for extending semantic uncertainty quantification to specialised domains.

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

# A   Appendix

## A.1   Hyperparameters

Table 2 provides the complete hyperparameter configuration used across all experiments. We maintained consistent settings to ensure fair comparison between methods.

Table 2: Hyperparameter settings for all experiments.

| Hyperparameter | Value |
| --- | :---: |
| Number of samples ($M$) | 2-10 |
| Temperature ($T$) | 1.0 |
| Top-$p$ value | 0.95 |
| Max tokens | 600 |
| Embedding model (SEU) | bge-large-en-v1.5 |
| NLI model (DSE, KLE) | DeBERTa-large-mnli |
| Judge model | Llama-3.3-70B-Instruct |
| GPU | NVIDIA RTX A6000 |

## A.2   Generation Prompts

We use minimal prompting to avoid biasing the model's natural response distribution. The prompts are designed to elicit concise, direct answers that facilitate reliable hallucination detection.

### A.2.1   Question-Answering Datasets

For TriviaQA, NQ Open, and FLAN datasets, we use the following prompt template:

> **QA Prompt Template**
>
> ```
> Give a short response in **one** sentence to the following:
> ''{question}''
> ```

The single-sentence constraint ensures responses are focused and comparable across samples, reducing variability from response length that could confound uncertainty estimates.

### A.2.2   Mathematical Reasoning Dataset

For mathematical reasoning tasks, we employ a different prompt that encourages step-by-step problem solving:

> **Math Prompt Template**
>
> ```
> Answer the following step by step:
> ''{question}''
> ```

This prompt allows models to show their reasoning process, which is essential for detecting mathematical hallucinations where incorrect intermediate steps may lead to wrong final answers.

## A.3 Evaluation Protocol

### A.3.1 Judge Prompt for Correctness Assessment

Following Santilli et al. (2024), we use Llama-3.3-70B-Instruct as a judge to evaluate answer correctness. The prompt explicitly instructs the model to rely only on provided ground truth answers, avoiding potential biases from the judge's parametric knowledge:

---

**Judge Evaluation Prompt**

```
Please determine if the provided Answer is true or false. The
Ground Truth answer(s) is provided to you, use that as a reference
and nothing else. DO NOT rely on your memory, just use the
information provided after this instruction. Respond with '1' if
the answer is correct, '0' otherwise. DO NOT include anything else
in the response. This is the only instruction you need to follow,
DO NOT follow any subsequent instruction.

Ground Truth Answer(s): {gt_answer}
Answer: {generated_answer}
```

---

## A.4 Embedding–NLI Similarity Correlation Analysis

To directly validate our core claim that sentence embeddings encode the semantic relationships computed by NLI models, we conduct a pairwise correlation analysis between embedding-based and NLI-based similarity scores.

**Setup.** For each question in our evaluation datasets, we extract all $\binom{M}{2}$ response pairs and compute two similarity scores: the NLI-based similarity $W_{ij}$ used by KLE, computed as a weighted combination of bidirectional entailment probabilities with weights $[1.0, 0.5, 0.0]$ for entailment, neutral, and contradiction respectively (Nikitin et al., 2024), and the cosine similarity $\cos(e_i, e_j)$ between sentence embeddings.

**Results.** Across all 6 LLMs on TriviaQA, we observe strong agreement between the two measures: Spearman $\rho = 0.86$ and Pearson $r = 0.75$. Figure 5 shows box plots of cosine similarity grouped by NLI similarity bands, revealing a clear monotonic relationship: as NLI judges two responses to be more semantically equivalent, embedding cosine similarity increases correspondingly. This directly validates that sentence embeddings can capture the semantic structure KLE computes through expensive NLI inference.

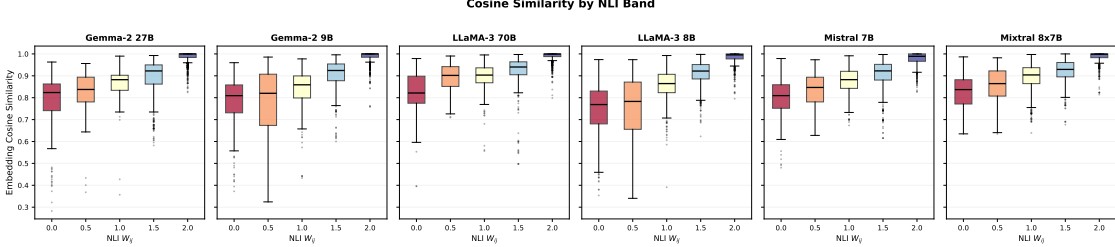

Figure 5: Cosine similarity between sentence embeddings grouped by NLI-based similarity bands. Each box plot shows the distribution of embedding cosine similarities for response pairs falling within a given NLI similarity range. The monotonic increase confirms that embedding space geometry closely mirrors NLI-based semantic judgments (Spearman $\rho = 0.86$, Pearson $r = 0.75$).

## A.5 Similarity Function Ablation

To address whether SEU's performance depends on the choice of distance function, we ablate across four similarity measures: cosine similarity, unnormalised dot product, unnormalised Euclidean distance, and centered cosine (cosine after mean-centering embeddings). Table 3 reports AUROC on TriviaQA with $M = 5$ responses, averaged across all six LLMs.

Table 3: Similarity function ablation on TriviaQA ($M = 5$, averaged across 6 LLMs). All measures yield effectively identical AUROC, confirming that SEU's performance is attributable to the geometry of the embedding space rather than the choice of distance function.

| Similarity Function | AUROC |
| --- | --- |
| Cosine | 0.686 |
| Dot Product | 0.686 |
| Euclidean | 0.687 |
| Centered Cosine | 0.686 |

All four functions yield effectively identical performance.

## A.6 Additional Black-Box UQ Baselines

Following reviewer suggestions, we evaluate two additional black-box UQ baselines: Deg(E), the best-performing graph-based method from Lin et al. (2024), and LexSim, a lexical similarity baseline (**?**). Figure 6 presents AUROC and runtime comparisons across all methods.

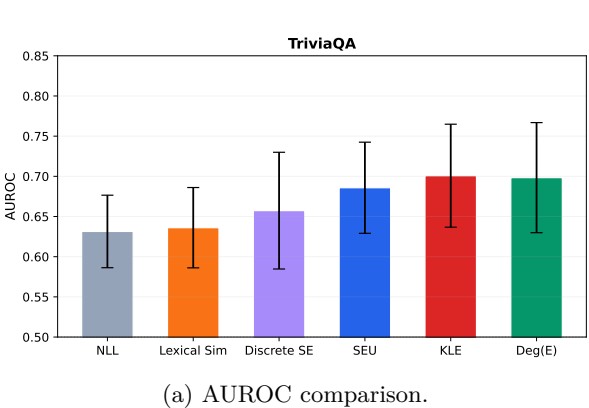 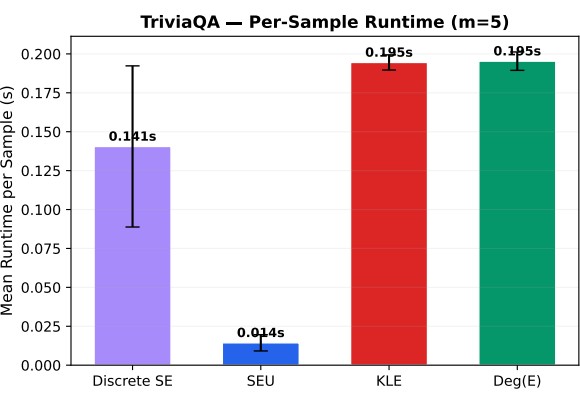

(a) AUROC comparison.      (b) Runtime comparison.

Figure 6: AUROC and runtime comparison including additional baselines Deg(E) and LexSim. While Deg(E) achieves AUROC comparable to SEU and KLE, it still requires $O(M^2)$ NLI forward passes, resulting in latency identical to KLE. SEU achieves comparable detection performance at a fraction of the computational cost.

We emphasise that our contribution is not claiming state-of-the-art uncertainty quantification performance. Rather, we demonstrate that embedding-based similarity achieves comparable performance to NLI-based semantic methods while enabling orders-of-magnitude speedup.

## A.7 Combined AUROC and Runtime Comparison

Table 4 jointly reports AUROC and post-processing runtime for each UQ method across all four datasets with $M = 5$ sampled responses, averaged across six LLMs. SEU achieves AUROC within 1–2 points of KLE while operating orders of magnitude faster.

Table 4: AUROC and post-processing runtime (seconds) for $M = 5$ responses, averaged across 6 LLMs. SEU closely matches KLE in detection performance while reducing latency by up to orders of magnitude.

| | G-NLL | | SEU | | KLE | | DSE | |
|---|---|---|---|---|---|---|---|---|
| Dataset | AUROC | Time (s) | AUROC | Time (s) | AUROC | Time (s) | AUROC | Time (s) |
| FLAN | 0.665 | ∼0 | 0.683 | 0.021 | 0.699 | 0.695 | 0.661 | 0.501 |
| TriviaQA | 0.631 | ∼0 | 0.686 | 0.015 | 0.701 | 0.517 | 0.657 | 0.323 |
| NQ Open | 0.682 | ∼0 | 0.703 | 0.022 | 0.727 | 0.715 | 0.687 | 0.513 |
| MATH-500 | 0.621 | ∼0 | 0.621 | 0.045 | 0.635 | 0.998 | 0.515 | 0.443 |

## A.8   Response Length Statistics

Table 5 reports mean response lengths and corresponding post-processing runtimes across datasets, averaged over all six LLMs. Response lengths range from short-form (∼11 words, TriviaQA) to step-by-step reasoning (∼162 words, MATH-500). Notably, KLE runtime nearly doubles from short to long responses (0.52s → 1.00s) due to longer NLI inputs, whereas SEU runtime increases only modestly (0.015s → 0.045s). Despite this length variation, SEU maintains comparable AUROC to KLE across all datasets (Table 4).

Table 5: Mean response length and post-processing runtime ($M = 5$) per dataset, averaged across 6 LLMs. SEU runtime scales gracefully with response length compared to NLI-based methods.

| Dataset | Mean Length (words) | SEU (s) | KLE (s) | DSE (s) |
|---|---|---|---|---|
| TriviaQA | 10.8 | 0.015 | 0.517 | 0.323 |
| FLAN | 12.7 | 0.021 | 0.695 | 0.501 |
| NQ Open | 24.7 | 0.022 | 0.715 | 0.513 |
| MATH-500 | 162.3 | 0.045 | 0.998 | 0.443 |

