# OpenReview forum: "Improving Uncertainty Quantification in Large Language Models via Semantic Embeddings"
_TMLR — Rejected by TMLR_

### Review · Reviewer_nhp3 · 2025-10-20

**Summary Of Contributions:**

The paper investigates uncertainty modeling in LLMs. The motivation primarily considers the computational complexity of current methods for quantifying such uncertainties. Furthermore, the authors highlight that these methods should be applicable to black-box APIs where token probabilities are inaccessible, while also capturing continuous semantic representations as done in KLE.

The core idea is employing an embedding space that inherently encodes semantic relationships. The authors propose Semantic Embedding Uncertainty, a metric that embeds each response and computes similarities via cosine similarity. This way, they replace time-intensive forward passes. The approach is faster than related methods while maintaining competitive performance in uncertainty quantification.

The proposed metric is evaluated against four baselines using AUROC to detect hallucinations. The experiments are quite extensive, with six LLMs across four datasets. Results demonstrate that their approach achieves lower computational complexity while maintaining reasonable performance. Additionally, the authors conduct a sensitivity analysis examining both the impact of NLI inference steps and the choice of embedding model.

**Audience:**

Yes

**Audience Explanation:**

This paper demonstrates that embedding-based approaches can achieve reasonable uncertainty quantification performance compared to state-of-the-art NLI-based methods while reducing computational complexity. I believe it can be interesting if the main claims are supported by experiments, as it can have a strong real-world impact.

**Claims And Evidence:**

No

**Claims Explanation:**

**Strengths:**

- **Clear Motivation:** The paper nicely motivates the problem of semantic uncertainty quantification as well as the computational bottleneck it tries to address.
- **Practical Impact:** The proposed solution can have a considerable impact as it provides a strong speedup compared to baselines.
- **Extensive Experiments:** The approach is evaluated on a lot of experimental settings with four baselines, six LLMs, and four datasets.

**Weaknesses:**

- **Insufficient evidence of the core claim:** The most critical problem I see is the analysis of the claims made by the authors. Specifically, the paper claims that "the embedding space already encodes the semantic relationships that KLE uses an NLI model to compute" (p. 3). However, this is not really validated directly, but rather through the benchmarks. In particular, when making this claim, the authors should analyze how embedding distances (or similarity such as cosine similarity) relate to the bidirectional entailments computed by NLI models. This is my biggest concern, as this relationship really shows whether the proposed SEU works because of the mentioned claim. Furthermore, it makes it far more intuitive to understand how SEU will behave in different cases, e.g., how good does the embedding need to be to allow SEU to reliably assess uncertainty?
I'd suggest analyzing this interaction and including an analysis in the sensitivity analysis.
- **Insufficient embedding model analysis:** Related to my first concern, the paper's analysis of embedding model selection is superficial, given its importance to SEU's performance. Despite the embedding model being a key component for computing the SEU scores, only a brief paragraph addresses this issue, and just two well-working embedding models are compared. However, what happens if the embedding model works poorly in a particular domain? How does uncertainty estimation degrade as embedding quality decreases? The paper provides no guidance on selecting appropriate embeddings for new domains, nor does it provide an analysis with low-quality or domain-mismatched models.
I suggest expanding the evaluation to include a bigger variety of embedding models, especially those with varying embedding quality. Additionally, the authors should provide principled criteria or heuristics for embedding model selection based on the domain. Without this, practitioners lack guidance for deploying SEU effectively, considerably limiting the method's practical applicability.
- **Limited novelty:** While the key insight (replacing NLI with embedding-based similarity) is relatively straightforward, the paper's contribution lies primarily in demonstrating its practical effectiveness for uncertainty quantification. The contribution is more empirically-focused than conceptually novel, which is okay given the bottleneck it addresses. However, the lack of analysis of the claims (as noted above) prevents the work from offering deeper insights beyond the empirical observation. Including such analyses would considerably improve the novelty of the paper.
- **(Minor) Representation of the paper:** While the quality of writing is generally clear, there are some redundancies in the text (e.g., explaining the bottleneck repeatedly) or not using TeX math symbols.
- **What time is measured:** For timing experiments, what is actually measured. This is rather vaguely explained in the main text and needs more clarification.

**Requested Changes:**

See Explanations regarding Claims.

---

> ### Author Response · Authors · 2026-02-12
> **Rebuttal to Reviewer nhp3**
>
> We thank the reviewer for their detailed and constructive feedback. The suggested analyses, particularly the direct correlation validation analysis, have substantially strengthened the paper. We address each point below.
>
> **On Weakness 1: Insufficient Evidence of the Core Claim**
>
> We thank the reviewer for this valuable suggestion. We agree that direct validation of the core claim strengthens the paper significantly. To this end, we now include a pairwise analysis comparing embedding similarity to NLI-based similarity. For each question, we extract all response pairs and compute both:
>
> - The NLI-based pairwise similarity $W_{ij}$ used by KLE, computed from bidirectional entailment scores with weights $[1.0, 0.5, 0.0]$ for entailment, neutral, and contradiction respectively, following the original KLE paper (Nikitin et al., 2024).
> - The cosine similarity of sentence embeddings.
>
> Across all 6 LLMs, we observe strong agreement: Spearman $\rho = 0.86$, Pearson $r = 0.75$. Figure 5 (Appendix A.4) visualises this relationship via box plots of cosine similarity grouped by NLI similarity band, demonstrating a clear monotonic relationship: as NLI judges two responses to be more semantically equivalent, embedding cosine similarity increases correspondingly. This directly validates that embeddings can capture the semantic structure KLE computes through expensive NLI inference.
>
> **On Weakness 2: Insufficient Embedding Model Analysis**
>
> **Symmetry with NLI model selection.** We note that KLE and SE face an analogous model selection problem: the choice of NLI model similarly affects their performance, yet prior work does not exhaustively analyse NLI model degradation across domains. Both methods have a swappable component requiring domain-appropriate selection.
>
> **Existing infrastructure for selection.** Sentence embeddings particularly benefit from well-established benchmarking infrastructure. The MTEB leaderboard (Muennighoff et al., 2022; https://huggingface.co/spaces/mteb/leaderboard) provides comprehensive quality scores across diverse domains, enabling practitioners to select embeddings with strong performance on tasks resembling their target domain. Additionally, our newly added correlation analysis (Spearman $\rho = 0.86$) provides a principled selection method: practitioners can compute this correlation on a small held-out sample to verify embedding suitability before deployment.
>
> **Scope of evaluation.** Exhaustive evaluation across all embedding quality levels, while informative, tests the embedding models themselves rather than SEU's contribution. Such an evaluation would be infeasible to conduct comprehensively given the rapidly growing landscape of embedding models and the time required to do each experiment. Our claim is that *when embeddings capture semantic similarity*, SEU approximates KLE efficiently, not that all embeddings work universally. This assumption holds for well-established embedding models with strong MTEB performance, which is the realistic deployment scenario. We have made this assumption explicit in Section 3.1 and provide the above selection guidance so practitioners can validate applicability before deployment.
>
> **On Weakness 3: Limited Novelty**
>
> Our newly added correlation analysis (Spearman $\rho = 0.86$, Pearson $r = 0.75$; Figure 5, Appendix A.4) directly validates *why* SEU works: modern sentence embeddings can capture the semantic similarity structure that NLI models compute through expensive bidirectional entailment.
>
> This insight extends beyond SEU. It suggests that for tasks relying on pairwise semantic comparison, practitioners may substitute NLI inference with embedding similarity more broadly. Combined with the explicit assumption statement (Section 3.1) and practitioner guidance for embedding selection (via MTEB and held-out correlation checks), we believe the revised paper offers both empirical and conceptual contributions to the uncertainty quantification community.
>
> **On Weakness 4: What Time is Measured**
>
> We thank the reviewer for highlighting this ambiguity. To clarify, we measure the *post-hoc overhead* of each UQ method, i.e., the time required to transform already-sampled responses into an uncertainty score. This is the correct comparison because the sampling cost is shared: all semantic methods (SE, KLE, DSE, SEU) require generating $M$ responses from the LLM, and this cost is identical across methods. The distinction is already noted in the Figure 1 caption ("average runtime for post-processing the generated responses"). We have also updated Section 4.2 to make this more explicit.

---

### Review · Reviewer_emPo · 2025-12-22

**Summary Of Contributions:**

The paper introduces Semantic Embedding Uncertainty (SEU), a computationally efficient method for quantifying semantic uncertainty in large language model outputs to detect hallucinations. Instead of relying on expensive pairwise natural language inference (NLI) comparisons as in prior approaches such as Semantic Entropy (SE), Discrete Semantic Entropy (DSE), and Kernel Language Entropy (KLE), SEU measures disagreement directly in sentence embedding space by computing average pairwise cosine distances among sampled responses. Experiments across multiple model families and benchmarks show that SEU achieves hallucination detection performance close to KLE while offering up to 100× lower latency, making it practical for real-time and resource-constrained settings.

**Audience:**

No

**Audience Explanation:**

I think technical novelty is limited. The work directly addresses a well-recognized bottleneck in semantic uncertainty methods and provides clear empirical evidence that a much cheaper embedding-based alternative can recover most of the performance at drastically lower latency . For researchers and practitioners concerned with real-time hallucination detection and systems-level efficiency, the paper offers a concrete and immediately usable takeaway: that sentence embeddings are often “good enough” substitutes for NLI-based semantic comparisons in uncertainty estimation.

That said, the interest is likely narrowly scoped. The contribution will resonate less with readers seeking new theoretical insights into uncertainty, semantic modeling, or learning objectives, since the method largely repurposes existing embedding techniques without introducing new modeling principles. Thus, while the paper may not appeal broadly to the full TMLR audience, it is likely of interest to a subset of them.

**Broader Impact Concerns:**

No outstanding ethical implications.

**Claims And Evidence:**

Yes

**Claims Explanation:**

Overall, the paper’s empirical claims are largely supported by accurate, clear, and convincing evidence, but the strength of the evidence mainly justifies a modest contribution rather than a strong one. The authors clearly demonstrate that SEU achieves AUROC performance close to KLE while dramatically reducing runtime, and these claims are consistently backed by quantitative results and ablations. However, the technical contribution itself is limited: SEU is essentially a straightforward replacement of NLI-based semantic similarity with off-the-shelf sentence embedding cosine distances. As a result, while the evidence supports the efficiency–performance trade-off claims, it does not substantiate any deeper algorithmic or theoretical advance in uncertainty quantification. I believe some members of the TMLR audience would find the findings useful, but the work falls closer to an engineering simplification than a conceptual innovation, which limits its overall significance despite solid empirical validation.

**Requested Changes:**

I find the justification in Section 3.2.1 for using cosine similarity insufficiently convincing. The authors mainly argue that cosine similarity is appropriate because modern embedding models are trained with contrastive objectives that optimize cosine similarity, and because benchmarks such as MTEB adopt cosine similarity as a standard evaluation metric. While these points are reasonable, they remain largely motivational rather than evidential. I would recommend strengthening this section with more quantitative analysis, such as empirical comparisons against alternative similarity measures (e.g., dot product, Euclidean distance, or centered cosine), or controlled experiments demonstrating that cosine similarity is particularly well aligned with semantic disagreement in the context of uncertainty estimation. Such analysis would significantly improve the rigor of the design rationale.

---

> ### Author Response · Authors · 2026-02-12
> **Rebuttal to Reviewer emPo**
>
> **On Cosine Similarity Justification**
>
> We thank the reviewer for this suggestion. Our use of cosine similarity supported by (i) a mathematical equivalence result under L2 normalisation, and newly added (ii) empirical verification (Table 3, Appendix A.5).
>
> **1. For L2-normalised embeddings, cosine, dot product, and Euclidean distance are rank-equivalent.** In our pipeline we L2-normalise embeddings before scoring, following standard usage for these models. For unit-norm vectors, dot product equals cosine similarity exactly, and squared Euclidean distance satisfies $\|a - b\|^2 = 2(1 - \cos(a, b))$. These three metrics therefore induce identical rankings, which means any rank-based evaluation metric (including AUROC) yields numerically identical results. To test whether explicit normalisation is necessary, **Table 3 (Appendix A.5)** reports AUROC using *unnormalised* dot product and Euclidean distance alongside centred cosine similarity. Emperically, all yield effectively identical scores to cosine similarity, demonstrating that SEU is robust to the choice of similarity function.
>
> **2. Prior empirical evidence.** Consistent with the above, Reimers & Gurevych (2019) [1] report that negative Manhattan and negative Euclidean distances yield roughly identical results to cosine similarity across all STS benchmarks. The MTEB benchmark [2] similarly adopts cosine-based Spearman correlation as its primary STS metric.
>
> **On Contribution Scope and Audience Interest**
>
> We appreciate the reviewer's acknowledgment that "researchers and practitioners concerned with real-time hallucination detection and systems-level efficiency" would find the takeaway "concrete and immediately usable."
>
> Our revised submission includes a correlation analysis (Figure 5, Appendix A.4) demonstrating that embedding cosine similarity strongly agrees with NLI-based similarity (Spearman $\rho = 0.86$, Pearson $r = 0.75$). This transforms the contribution from an engineering simplification to a validated insight, modern sentence embeddings can capture the same semantic structure that NLI models compute through expensive bidirectional entailment. This finding is transferable beyond SEU to tasks relying on pairwise semantic comparison.
>
> TMLR welcomes empirical studies and applications of existing methods that yield new insight or clarify strengths and weaknesses, and does not require methodological novelty for acceptance. We believe our work falls well within this scope: we provide new empirical insight into the relationship between embedding similarity and NLI-based similarity, and demonstrate that this insight enables a $100\times$ speedup for semantic uncertainty quantification without meaningful loss in detection performance.
>
> **References:**
> - [1]Reimers & Gurevych, 2019. "Sentence-BERT: Sentence Embeddings using Siamese BERT-Networks"
> - [2] Muennighoff et al., 2022. "MTEB: Massive Text Embedding Benchmark"

---

### Review · Reviewer_XCDA · 2026-01-14

**Summary Of Contributions:**

The authors propose Semantic Embedding Uncertainty (SEU), a black-box method for computing uncertainty scores based on multiple generations for a single prompt. The approach involves embedding these generations using an external encoder and calculating a score based on their pairwise cosine similarity. The primary advantage of SEU over existing methods is its significantly reduced latency. Experimental results demonstrate that the approach achieves performance comparable to existing baselines in terms of AUROC while being substantially faster at computing uncertainty scores.

## Strengths

- **Efficiency in UQ:** The paper addresses the critical problem of improving the efficiency of black-box uncertainty quantification (UQ) methods. While approaches such as Semantic Entropy (SE) and Kernel Language Entropy (KLE) have shown promising results, they remain computationally expensive. This work provides a viable path forward in this direction, which is vital given that uncertainty remains a major bottleneck for LLM safety.

- **Writing Quality:** The paper is well-structured and clearly written.

- **Rigorous Hallucination Evaluation:** The evaluation of hallucination detection is well-executed. The authors employ standard metrics and adhere to best practices established in current literature.

- **Model Diversity:** Evaluation was performed across six different LLMs, including Gemma 2 (9B/27B), Llama 3 (8B/70B), Mistral 7B, and Mixtral 8×7B, ensuring the findings generalize across different architectures.


## Weaknesses

- **Limited black-box UQ Baseline Comparison:** The method is only evaluated against two primary black-box semantic methods (SE and KLE). Consequently, the performance of SEU in terms of AUROC and latency relative to a broader range of black-box methods is unclear. I recommend including other UQ baselines such as:
    - Verbalized uncertainty (utilizing KV cache) [0]
    - Number of semantic sets [1]
    - Sum of eigenvalues of the graph Laplacian [1]
    - Degree matrix (Deg) [1]
    - Eccentricity (Ecc) [1]
    - Lexical similarity (LexSim) [2]

- **Focus on Short-Form Generation:** The evaluation is primarily performed on short-form generation. It is unclear whether the method remains effective for long-form generation. To address this, the authors should conduct experiments using prompts that elicit long-form responses, as seen in the original Semantic Entropy paper [3]. Currently, the only dataset involving potentially longer reasoning is MATH-500 (but it's not clear since length is not reported).

- **Dataset Diversity and Domain Claims:** I believe the evaluation should be extended to other datasets. Following [3], the authors could include BioASQ, SQuAD (with context), and SVAMP; SimpleQA [4] would also be a valuable addition. Furthermore, in the paper, you claim that "SEU also extends naturally to specialised domains through model selection. Medical applications can employ BioBERT..." and further mention finance and legal settings. However, there is no empirical evidence provided to support the claim that the method is effective in these specific domains.

- **Impact of Response Length:** It is not clear how generated answer length impacts SEU’s AUROC and latency performance. Embedding modules may be less performant on longer generations, but this is difficult to assess without empirical comparison. For instance, [3] used an LLM to check entailment to sidestep this. This is an important aspect that appears overlooked and may impact performance [5].

- **Methodological Rigor:** The method is relatively simple and relies on intuition rather than a rigorous mathematical foundation. The authors propose an uncertainty score rather than a principled estimator, and its mechanism is largely based on heuristics. For instance, the selection of cosine similarity is debatable and not clearly justified. While the authors note that “cosine similarity is a dominant metric for semantic comparison benchmarks,” it is difficult to defend it as the definitive choice. Prior work [5] has shown that embedding-based metrics such as SentenceBERT can fail to accurately capture similarity when using cosine similarity, leaving the impact of this limitation on SEU unclear.

- **Latency Evaluation Rigor:** The evaluation of latency is less rigorous than the AUROC evaluation:
    - Although latency is the core contribution, results for latency are only presented in Figure 1 and are limited to three datasets. Why is the latency for MATH-500 omitted?
    - The presentation in Table 1 is suboptimal. Latency and AUROC should be displayed together to allow for a direct comparison of the trade-offs. Furthermore, switching between Figure 1 and Table 1 makes this comparison difficult.
    - The embedder model and the NLI model have different parameter counts and possibly inference costs. This discrepancy should be more explicitly accounted for when calculating relative latency gains.

[0] Xiong et al., 2023 "Can LLMs Express Their Uncertainty? An Empirical Evaluation of Confidence Elicitation in LLMs"

[1] Lin et al., 2023 "Generating with Confidence: Uncertainty Quantification for Black-box Large Language Models", TMLR

[2] Fomicheva et al., 2020 "Unsupervised quality estimation for neural machine translation"

[3] Farquhar et al., 2024 "Detecting hallucinations in large language models using semantic entropy", Nature

[4] Wei et al. "Measuring short-form factuality in large language models."

[5] Santilli et al. "Revisiting uncertainty quantification evaluation in language models: Spurious interactions with response length bias results."

**Audience:**

Yes

**Audience Explanation:**

As detailed in the Strengths section, efficient and reliable uncertainty quantification for LLMs is an important open problem.

**Claims And Evidence:**

No

**Claims Explanation:**

As detailed in the Weaknesses section, I believe that to claim the superiority of SEU, the paper requires additional UQ baselines, a more diverse set of datasets (particularly for long-form settings), and a more comprehensive evaluation for latency.

**Requested Changes:**

- Integrate additional UQ baselines for comparison.
- Expand the evaluation to include more diverse datasets.
- Include experiments specifically targeting the long-form generation setting.
- Analyze the impact of generation length on the performance and reliability of the embedding module.
- Improve the presentation of results, specifically by consolidating latency and performance metrics to highlight the primary efficiency claims.

---

> ### Author Response · Authors · 2026-02-12
> **Rebuttal to Reviewer XCDA**
>
> We thank the reviewer for their detailed feedback, which has helped us strengthen the paper. We have now made revisions to address the concerns.
>
> **On Weakness 1: Limited Black-Box UQ Baseline Comparison**
>
> We have added two baselines from the suggested list:
>
> - **Deg(E)**: The best-performing method from Lin et al. (2023)
> - **LexSim**: Lexical similarity baseline (Fomicheva et al., 2020)
>
> Results (Figure 6, Appendix A.6): While Deg(E) performs comparably to SEU and KLE in AUROC, it still requires the $O(M^2)$ NLI forward passes to construct the similarity matrix, resulting in identical latency to KLE while marginally underperforming.
>
> Regarding the **number of semantic sets** (Kuhn et al., 2023): this metric counts the number of clusters formed by bidirectional entailment and is therefore very similar to DSE without the entropy computation and both rely on the same NLI-based clustering. DSE already serves as this baseline in our experiments.
>
> We emphasize that our contribution is not claiming SOTA uncertainty quantification performance. Rather, we demonstrate that embedding-based similarity achieves comparable performance to NLI-based semantic methods while enabling orders of magnitude faster processing.
>
> **On Weakness 2: Focus on Short-Form Generation**
>
> Our evaluation includes MATH-500, where responses contain step-by-step reasoning with average length $\sim$162 words across 6 LLMs (Table 4, Appendix A.7). SEU performs comparably to KLE on this dataset, demonstrating robustness to longer generations. We additionally report response length statistics across all datasets in Table 5, showing that SEU's performance and runtime scale gracefully from short-form ($\sim$11 words, TriviaQA) to moderate-length generations ($\sim$162 words, MATH-500).
>
> **On Weakness 3: Dataset Diversity and Domain Claims**
>
> We acknowledge this criticism is valid. The statements regarding domain-specific embeddings (BioBERT, FinBERT, LEGAL-BERT) were speculative. We have revised Section 3.2.1 and Section 6 to frame this as a potential advantage of the modular architecture rather than a demonstrated capability, explicitly noting that empirical validation in specialised domains is left for future work.
>
> **On Weakness 4: Impact of Response Length**
>
> Response length statistics and their impact on runtime are reported in Table 5 (Appendix A.8). Notably, KLE runtime nearly doubles from short-form to long-form responses (0.52s $\to$ 1.00s), whereas SEU runtime increases only modestly (0.015s $\to$ 0.045s), maintaining an order-of-magnitude faster processing speed across all response lengths. SEU also maintains comparable AUROC to KLE across this range (Table 4), indicating that embedding quality does not degrade significantly for longer length generations.
>
> **On Weakness 5: Methodological Rigor**
>
> We agree that direct validation of the core claim strengthens the paper significantly. To this end, we now include a pairwise analysis comparing embedding similarity to NLI-based similarity. For each question, we extract all response pairs and compute both:
>
> - The NLI-based pairwise similarity $W_{ij}$ used by KLE, computed from bidirectional entailment scores with weights $[1.0, 0.5, 0.0]$ for entailment, neutral, and contradiction respectively, following the original KLE paper (Nikitin et al., 2024).
> - The cosine similarity of sentence embeddings.
>
> Across all 6 LLMs, we observe strong agreement: Spearman $\rho = 0.86$, Pearson $r = 0.75$. Figure 5 (Appendix A.4) visualises this relationship via box plots of cosine similarity grouped by NLI similarity band, demonstrating a clear monotonic relationship: as NLI judges two responses to be more semantically equivalent, embedding cosine similarity increases correspondingly. This directly validates that embeddings can capture the semantic structure KLE computes through expensive NLI inference.
>
> **On Weakness 6: Latency Evaluation Rigor**
>
> We have addressed all three sub-concerns:
>
> **MATH-500 latency:** Now included in Table 4 alongside all other datasets. MATH-500 exhibits the largest runtime gap between methods, KLE takes $\sim$1.0s per evaluation compared to SEU's $\sim$0.045s, consistent with longer responses requiring more expensive NLI forward passes. This further reinforces SEU's advantage in latency-sensitive settings.
>
> **Presentation:** Table 4 now displays AUROC and runtime side-by-side across all four datasets for direct trade-off comparison, as requested. Additionally, Table 5 (Appendix A.8) reports how runtime scales with response length, providing a comprehensive view of the efficiency–performance trade-off.
>
> **Parameter counts:** The embedding model (335M parameters) and NLI model (406M parameters) are comparable in size, as noted in Section 4. The $100\times$ speedup is therefore not attributable to model size asymmetry but to a fundamental difference in computational complexity: SEU requires $O(M)$ forward passes, whereas KLE and DSE require $O(M^2)$ forward passes.

---

### Decision · Action_Editor_inUL · 2026-05-31

**Recommendation:** Reject

**Additional Comments:**

I thank the authors for their efforts in preparing a thorough rebuttal and for making several meaningful revisions to the manuscript. However, all the three reviewers provided a reject/leaning to reject recommendation. Based on the same, after carefully considering the reviews and responses, I am recommending rejection with an invitation to submit a major revision.

The authors are recommended to revisit the reviews and provide a substantial revision that addresses the all the points, including:
Providing sufficient evidence for the core claims.
Additional UQ baselines: A verbalized uncertainty baseline that leverages the KV cache should be included for a fair comparison.
Long-form generation: Only one long-form experiment is presented, which is insufficient to assess SEU in this setting. More comprehensive evaluation is needed.
Additional datasets: The study should include BioASQ, SQuAD with context, SVAMP, and SimpleQA to better demonstrate generalization.
Evaluation rigor: The experiments section should be reorganized to clearly analyze whether SEU is optimal on the latency–AUROC Pareto frontier.

**Audience:**

Yes

**Audience Explanation:**

Efficient and reliable uncertainty quantification for LLMs is an important open problem with clear practical relevance. The paper addresses a well-recognized computational bottleneck in black-box semantic UQ methods and demonstrates that embedding-based similarity can approximate NLI-based methods at dramatically lower latency. While the audience is likely narrowly scoped — primarily practitioners and researchers focused on real-time hallucination detection rather than those seeking theoretical advances — this subset of the TMLR readership would find the findings concrete and actionable.

**Claims And Evidence:**

No

**Claims Explanation:**

The paper's central efficiency claim — that SEU achieves comparable hallucination detection performance to NLI-based methods at dramatically lower latency — is reasonably well supported by the existing experiments across six LLMs and four datasets. The newly added correlation analysis between embedding cosine similarity and NLI-based pairwise similarity (Spearman and Pearson agreement) also meaningfully strengthens the theoretical grounding of the method.
However, the evidence remains insufficient to fully substantiate the claims in their current scope. The reviewers have pointed out that there are incomplete baseline comparisons and limited generalization evidence. The evaluation covers primarily short-form QA settings. The claim that SEU generalizes robustly to long-form generation remains inadequately supported. Standard benchmarks in this area such as BioASQ, SQuAD with context, and SVAMP have not been included despite being specifically requested by reviewers. The paper claims practical superiority in the latency–accuracy trade-off space, but does not present a systematic analysis of the same Without this framing, the efficiency advantage is asserted rather than demonstrated. Domain-specific claims have retracted but not replaced. The original submission made claims about SEU's applicability in medical, legal, and financial domains via domain-specific embeddings. The revised framing as a "potential advantage" is speculative and unsupported.
In summary, while the core empirical results are credible and directionally convincing, the evidence does not yet meet the standard required to support the full set of claims made in the paper. A more comprehensive evaluation addressing the points above is needed.

**Resubmission Of Major Revision:**

The authors may consider submitting a major revision at a later time.